# Postoperative Results After Patient Blood Management with Intravenous Iron Treatment Implementation for Preoperative Anemia: Prospective Cohort Study of 1294 Colorectal Cancer Patients

**DOI:** 10.3390/cancers17060912

**Published:** 2025-03-07

**Authors:** Ana Centeno, Carlos Jerico, Lana Bijelic, Carmen Deiros, Sebastiano Biondo, Jordi Castellví

**Affiliations:** 1Colorectal Surgery Unit, General Surgery Department, Complex Hospitalari Moisès Broggi—Consorci Sanitari Integral, 08970 Sant Joan Despí, Spain; 2Anemia Clinic, Internal Medicine Department, Complex Hospitalari Moisès Broggi—Consorci Sanitari Integral, 08970 Sant Joan Despí, Spain; 3Peritoneal Carcinomatosis Unit, General Surgery Department, Complex Hospitalari Moisès Broggi—Consorci Sanitari Integral, 08970 Sant Joan Despí, Spain; 4Anesthesiology and Resuscitation Department, Complex Hospitalari Moisès Broggi—Consorci Sanitari Integral, 08970 Sant Joan Despí, Spain; 5Colorectal Surgery Unit, General Surgery Department, Hospital Universitari de Bellvitge, 08907 L’Hospitalet de Llobregat, Spain

**Keywords:** anemia, colorectal cancer, intravenous iron, preoperative anemia, patient blood management, anemia clinic, postoperative results, complications

## Abstract

Preoperative Anemia (PA) is frequent in Colorectal Cancer (CRC) patients, and treatment is challenging due to continuous blood loss and proinflammatory status. Intravenous Iron (IVI) as part of a Patient Blood Management (PBM) protocol is a safe and effective measure prior to surgery and could reduce the need for Red Blood Cell Transfusion (RBCT). Although Enhanced Recovery After Surgery (ERAS) protocols and PBM recommendations focus on restrictive transfusion policies, barriers still exist that limit implementation of such protocols and systematic treatment within institutional Guidelines. This study analyses the impact of IVI treatment on Hemoglobin levels, surgical results, and RBCT rates, while being administered for PA in CRC within an institutional PBM pathway in a high-volume CRC Unit.

## 1. Introduction

Preoperative Anemia (PA) is frequent in newly diagnosed Colorectal Cancer (CRC) patients [1,2,3,4], and its treatment is challenging due to proinflammatory status and limited time until surgery [5,6,7,8]. Red Blood Cells Transfusion (RBCT) can lead to a rapid increase in Hemoglobin (Hb) levels but is associated with an increased risk for postoperative complications [1,2,9,10] and poorer results after surgery [11,12,13,14,15]. Pharmacological alternatives for PA treatment have been proposed [5], but their effects on postoperative results have not been unequivocally proven and are still a subject of debate [5,9,10,11].

Enhanced Recovery After Surgery (ERAS) pathways and Patient Blood Management (PBM) protocols focus on restrictive transfusion policies and PA assessment before surgery [1,16,17,18,19,20]. Current evidence supports the use of Intravenous Iron (IVI) in the preoperative setting due to its association with fast PA correction [10,12,20,21], however, there is still controversy regarding the efficacy in patients with a narrow preoperative time period and the optimal treatment strategy in the preoperative setting [5,22,23,24,25]. Consequently, barriers still exist that limit the implementation of protocol-based treatment algorithms in most CRC surgery programs [26,27,28].

The present study aims to evaluate postoperative results after systematic, prospective implementation of a PBM protocol that includes standardized preoperative iron-deficient PA treatment with IVI in a high-volume CRC Surgery Unit. We hypothesized that standardized preoperative IVI treatment for PA would decrease surgery-related complications, including the number of patients receiving RBCT.

## 2. Materials and Methods

PBM protocols were first introduced in our hospital in 2011 and expanded to include the CRC Surgery Unit in 2012 when prospective collection of data of all patients diagnosed with CRC and assessed in the PBM Anemia Clinic started.

This is an observational cohort study based on the prospectively collected data between January 2012 and December 2018 in our hospital, a 330-bed public inpatient facility serving a population of 425,000 inhabitants.

Analysis was conducted according to the STROBE [29] guidelines and registered in Clinical Trials (NCT06026618). Ethical approval was granted by CSI—Hospital Universitari de Bellvitge Research Ethics Committee (ref. no. CEIC HUB PR089/21 and CSI 21/19).

### 2.1. Perioperative Assessments

All patients were initially assessed by a colorectal surgeon and then referred to the PBM Anemia Clinic based on the results of their anemia screening panel. Patients were prescribed IVI supplementation (Ferric Carboxymaltose, 50 mg/mL CSL Vifor Saint Galene Switzerland^®^) if they presented with iron deficiency (Ferritin below 30 mg/mL or 30–300 mg/mL with a Transferrin Saturation below 20%) and a Hb value of less than 12 g/dL or between 12 and 13 g/dL with risk factors (high risk for bleeding, anticoagulant therapy, or less than 15 days until scheduled surgery). Dosage was estimated using a modified Ganzoni formula [30] (IVI dose = [(14 − Hb level) × 2.4 × Weight (kg) + 500] mg) 30–31 or the Simplified Strategy [31] for IVI Carboxymaltose, as it is shown in Figure 1.

Patients with mild iron-deficient PA (Hb 12–13 g/dL) who did not meet the criteria for referral to the PBM Clinic were recommended to receive oral iron and/or vitamins or folates (if indicated), though oral supplementation was not controlled by the PBM Clinic.

Treatment was to be completed at least 1 week before scheduled surgery, and patients had to be scheduled for surgery within 30 days of the initial assessment according to National standards for patient safety and quality in Cancer surgery.

All patients underwent surgery after multidisciplinary consensus and within an institutional ERAS program. Whenever possible, minimally invasive surgery was performed. After discharge, patients were scheduled for a postoperative visit within 30 days and henceforth proceeded with standard follow-up according to international recommendations.

The complete assessment and intervention protocol is visually summarized in Figure 1.

### 2.2. Study Groups

For the purpose of this study, patients were divided into three groups: non-anemic patients (Hb > 13 g/dL) were included in Group 1, mildly anemic patients (Hb 12–13 mg/dL) without criteria for IVI therapy were included in Group 2, and patients treated with IVI (iron deficiency and Hb < 12 mg/dL or Hb 12–13 mg/dL with risk factors) were included in Group 3.

Exclusion criteria were emergency surgery and treatment with RBCT before assessment in the PBM Clinic, although the latter was considered as a separate group and also included in a comparison with Group 3 patients. Patients diagnosed with non-iron-deficient anemia, patients receiving treatment out of protocol (non-anemic or non-iron-deficient patients receiving IVI outside of PBM pathways), or receiving treatment with other IVI preparations or oral iron (not controlled by the PBM clinic) were excluded from the effectiveness analysis.

An overview of patient allocation into study groups and excluded patients is shown in Figure 2.

### 2.3. Outcome Measures

The main objective of this study was to determine the effectiveness of the PBM pathway with IVI treatment measured by the degree of Hb change from baseline to Day of Surgery (DOS), to 24–48 h after surgery, and at discharge in patients treated with IVI and also by the differences in Hb values between groups. Treatment was considered effective based on a higher Hb level after receiving IVI and if changes remained significant until discharge.

Secondary objectives included an assessment of IVIs effect on CRC patients undergoing surgery in terms of surgery-related complications according to the Clavien-Dindo scale and the number of patients receiving RBCT between groups as well as Length of Stay (LOS).

Group 3 patients were also compared with patients receiving RBCT (and thus excluded from effectiveness analysis, as shown in Figure 2) in terms of Hb changes, RBCT rate, and complications.

It also analyzed the changes over time of good practice indicators such as RBCT rate (pre- and postoperatively), laboratory-based triggers (Hb values before RBCT), and the number of patients receiving IVI. Additionally, a correlation and multivariate analysis were performed with the intention of identifying possible predictors of complications, the need for RBCT, and LOS.

### 2.4. Statistical Analysis

Data were collected prospectively. Comparisons between groups were carried out using the Pearson Chi-squared test for categorical variables, ANOVA for normally distributed data, or the Kruskal–Wallis test for non-normally distributed data within continuous variables. *p*-values for pairwise comparisons were adjusted for multiplicity using Tukey’s correction. For the analysis of potential predictors of Complications, RBCT, and LOS, a correlation analysis was first performed with a Spearman test, and then a multivariate analysis with a linear regression model was performed, and estimated coefficients with 95% confidence intervals (CIs) were reported. Data were analyzed by complete case analysis, and missing values were not imputed. The significance level was set at 0.05 in all tests. Analysis was performed using Stata version 15.1 (StataCorp, 2017. Stata Statistical Software: Release 15. College Station, TX, USA: StataCorp LLC).

## 3. Results

Between January 2012 and December 2018, there were 1294 patients diagnosed and planned for surgery for primary CRC. Of these, 20 required emergency surgery, 93 received RBCT prior to evaluation (and were included in a separate group for subsequent analysis as specified), 30 were diagnosed with non-iron deficient anemia, and 19 and 15 received OI or Sucrose-based IVI (not controlled in the PBM Clinic) and were excluded. The remaining 1117 patients were eligible to enter the PBM pathway, were analyzed according to the previously described study groups, and underwent elective surgery for CRC.

There were 564 patients included in Group 1 (27 excluded for receiving IVI out of protocol), 141 in Group 2 (5 excluded for receiving IVI out of protocol), and 349 in Group 3 (26 excluded for not receiving IVI despite meeting criteria).

Patients who met the criteria for IVI (Group 3) were significantly older, more likely to be female, have hypertension, diabetes, and cardiomyopathy, and receive anticoagulant treatment compared to non-anemic patients. There were no significant differences between Group 1 and Group 2 patients in baseline characteristics, as there were also none between Group 3 patients and excluded patients receiving RBCT except in terms of Cardiopathy (29.0% vs. 50.5%, *p* = 0.00). The complete demographic and baseline characteristics of the cohort are displayed in Table 1.

Laparoscopic surgery was performed in 59.4% of patients in the whole cohort, with a statistically higher proportion in Group 1 patients (63.8%, *p* = 0.00). The most frequently performed surgeries were Right Colectomy (30.5%), Sigmoidectomy (28.5%), and Low Anterior Resection (25.6%). These frequencies varied along the Groups, with a higher proportion of Right Colectomy in Group 3 patients (47.3%, *p* = 0.00) and of both Sigmoidectomy and LAR in Group 1 patients (32.6% and 31.0%, respectively, *p* = 0.00).

### 3.1. Effectiveness of IVI Treatment

Mean baseline Hb levels were statistically significantly different between the 3 groups as expected: 14.5 g/dL (±1.0) in Group 1, 12.5 m/dL (±0.3) in Group 2, and 9.9 g/dL (±1.5) in Group 3 (*p* = 0.001). Iron deficiency was common in all patients regardless of their basal Hb level (67.6% of the whole cohort) and even higher in Group 3 patients (89.7%). Hb values and indicators of iron deficiency are shown in Table 2.

Group 3 patients presented with a significant increase in DOS Hb level after IVI treatment (11.1 g/dL compared to baseline 9.9 g/dL; *p* = 0.001).

A comparison between baseline and DOS Hb levels for all three groups shows an absolute decrease in Groups 1 and 2 (−0.6 g/dL or −4% and −0.4 g/dL or −3.2%, respectively) as opposed to an increase in Group 3 (1.3 g/dL or 15%). The difference between the groups remained significant, but the gap between the groups narrowed until discharge. Additionally, pairwise comparisons using Tukey’s range test showed no differences within Groups 2 and 3 patients in terms of Hb values at discharge (*p* = 0.265). Approximately 45.9% of all patients were discharged with Hb below 11 g/dL, even in the Group 1 patients in which that percentage reached 28.2%. The Hb change between baseline, DOS, 24–48 h after surgery, and discharge levels for all 3 groups is depicted in Figure 3.

The receipt of IVI treatment did not lead to a delay in surgical treatment, as evidenced by the number of days between diagnosis and surgery, which was not significantly different among the three groups (*p* > 0.05).

### 3.2. Surgical Results and Postoperative Complications

Complications were reported in 30.9% of cases, of which 6.5% were classified as severe (Clavien-Dindo III–IV). Mean LOS was 9.2 (±6.4) days, and 30-day Mortality was 1.4% in all the series, without differences among Groups.

Group 1 patients presented with statistically significantly fewer complications (*p* = 0.01) and were less likely to need RBCT (3.4%, *p* = 0.00), compared to Groups 2 and 3 patients (13.0% and 21.6%, respectively). Group 3 patients also presented with more complications regardless of severity (*p* < 0.05), mostly related to respiratory complications of heart failure (3.2% and 6.2%, respectively, *p* < 0.05), but pairwise comparisons using the Tukey’s range test showed no differences with Group 2 patients. There were no differences between groups in the incidence of anastomotic leak. A summary of complications is shown in Table 3.

### 3.3. Predictors of Complications

Variables such as Group, Age, Sex, Cardiopathy, American Society of Anesthesiology (ASA) scale, Neoadjuvant therapy, Baseline Hb value, DOS Hb value, Laparoscopy, and Type of Surgery performed were identified as potentially explicative variables for complications after surgery in a bivariate analysis (*p* < 0.05). The multivariate analysis showed that Male Sex (OR 1.36, IC95% 1.04–1.80), ASA ≥ III (OR 3.10, IC95% 1.25–7.73), and Neoadjuvant therapy (OR 2.05, IC95% 1.49–2.82) were independent factors for complications. The group was also kept in the model for adjustment, which showed that Group 3 patients had more risk of complications compared to those in Group 1 (OR 1.43, IC95% 1.06–1.93) and to those in Group 2 (OR 1.65, IC95% 1.09–2.50).

This analysis was also performed with complications classified according to the Clavien-Dindo scale. In this model, both Male Sex and Neoadjuvant therapy were independent factors only for Clavien-Dindo I-II complications (OR 1.36, IC95 1.01–1.83 and OR 1.60, IC95% 1.14–2.24, respectively). Also, Group 3 patients were not found to be an independent factor for Clavien-Dindo III-IV complications relative to Group 2 patients (OR 1.43, IC95% 0.64–3.19).

### 3.4. Patients Receiving RBCT Before PBC Clinic Assessment

Patients receiving preoperative RBCT (n = 93) presented with a mean Hb of 6.9 (±1.1) d/dL at the moment of diagnosis and received a mean of 1.97 (±0.80) units of RBCT before surgery. These patients’ mean DOS Hb values were 10.8 (±1.9) g/dL and 10.4 (±1.5) g/dL at discharge, which showed no difference from those of Group 3 patients’ (*p* = 0.14 and *p* = 0.81, respectively). However, the RBCT rate was 32.6%, which was significantly higher than that of Group 3 (21.6%, *p* = 0.026). Group 3 patients who received RBCT during the postoperative period needed a mean of 1.97 (±1.29) units compared to 1.8 (±1.2) units in the excluded patients (*p* = 0.89). There were no significant differences in terms of complications or LOS among these patients (*p* > 0.05).

### 3.5. Good Practice Indicators and Changes over Time

The postoperative RBCT rate was 10.7% during the years of the study, with a steady decrease over time until 5.9% in 2018. Adherence to optimal laboratory-based triggers for transfusion (Hb values below 7 g/dL or below 8 g/dL in patients with prior history of Cardiopathy) was 31.5% along the years of the study, with a clear trend towards improvement reaching 69.2% in 2018 and without statistically significant differences among the groups (*p* > 0.05). These results are depicted in Figure 4, as well as the total number of patients treated with IVI in the PBM Clinic over the years, which also showed an increase over the years, reaching up to n = 60 (39.7% of patients within the study) in 2018.

### 3.6. Multivariate Analysis

Finally, multivariate analyses were performed in order to identify potential predictors for complications, need for RBCT, and LOS. Variables such as Group, ASA and Comorbidities, Neoadjuvant therapy, Laparoscopy, and Type of Surgery performed were identified as potentially explicative variables in bivariate analysis and tested in three different independent models.

Even if Group was identified as a statistically significant factor in the LOS model (*p* = 0.03, Group 3 vs. Group 1, mean increase of 0.21 days IC95% 0.01–0.44), it was not a statistically significant variable for complications (*p* = 0.23) and need for RBCT (*p* = 0.45). ASA was a statistically significant predictor in all three models, and also variables identified as potential predictors were Sex (*p* = 0.02, Male vs. Female OR = 1.36 IC95% 1.04–1.8) for complications, DOS Hb values (*p* > 0.01, OR = 0.63 IC95% 0.52–0.73) for RBCT, and Laparoscopy (*p* < 0.01) and Type of Surgery (*p* < 0.01) for LOS.

## 4. Discussion

In this large cohort study, we evaluated the effect of IVI treatment on newly diagnosed primary CRC patients with iron-deficient PA. We showed that completion of IVI treatment was a feasible and effective measure, leading to a significant increase in Hb levels without delays to surgical resection, which was maintained during hospitalization days even after the unavoidable blood loss linked to surgical procedures. Furthermore, severely anemic patients treated with IVI (Group 3 patients) were discharged with Hb levels comparable to those in Group 2 (patients not eligible for IVI treatment), even though their baseline and DOS Hb levels were significantly lower.

While other trials, like PREVENTT, included patients undergoing major abdominal surgery and diagnosed with any form of anemia, in this study we focused solely on patients diagnosed with CCR and iron-deficient PA. This is of particular importance since worsening of PA, due to some degree of blood loss, is to be expected between diagnosis and the day of surgery. Our main result showing that IVI administration is able to increase Hb values during this period despite the continued blood loss proves its utility in CCR programs and could avoid the need for RBCT. Furthermore, changes in Hb values in Group 3 patients, however modest, should be interpreted keeping the tendency to a decrease shown in the other groups of the study in mind, which suggests an even greater increase linked to IVI treatment.

Prior studies, like the IVICA trial, did not find a reduction in RBCT associated with IVI treatment. In our cohort, we identified a target group (iron-deficient PA with Hb values < 12 g/dL) that can potentially benefit from IVI treatment due to a decrease in terms of RBCT in the postoperative period. Interestingly, correction of Hb values was comparable between severely anemic patients receiving RBCT in the preoperative period and those treated with IVI (Group 3 patients), but the RBCT rate in the postoperative setting was significantly lower in the IVI-treated patients, with a difference of more than 10%.

As stated, CRC patients are prone to a slow but steady decrease in Hb values during the preoperative period, but also after surgery. Anemic patients receiving IVI during the preoperative period showed a significantly lower decrease in Hb values, which suggests a protective effect after treatment linked to iron repositioning. This seemingly lasting effect of IVI treatment could be one of the factors determining a reduction in RBCT rate after surgery.

Patients in Group 3 were significantly older and had more comorbidities and a higher ASA score, and thus were expected to present with more complications during the postoperative period. Accordingly, they were found to have a higher risk for respiratory and cardiological complications, and multivariate analysis showed that variable Group 3 was an independent risk factor for Clavien-Dindo III-IV complications. However, we did not find differences in severe complications such as anastomotic leak, reinterventions, and mortality, or even LOS within the groups, and neither did we find a higher risk for Clavien-Dindo III-IV complications in Group 3 patients compared to those in Group 2. We believe that prehabilitation, early PA detection, and fast correction with IVI might reduce the negative effect of anemia itself on postoperative results, leading to rates of complications similar to Group 2. Moreover, we believe PA could be considered a frailty marker of paramount importance among elderly patients diagnosed with oncologic diseases, and its correction with IVI within a system-based PBM plan should be considered as a quality indicator in CCR programs.

Despite available literature, implementation of PBM protocols is still not the norm among patients undergoing oncologic surgery. Building multidisciplinary teams is a challenge as it incorporates PBM strategies but ensures a successful implementation and improves adherence. Our results show that standardized pathways with restrictive transfusion policies and protocolized IVI treatment can be implemented and expanded to a high-volume CRC surgery unit with satisfactory results and adherence among professionals. The Spanish National Health System offers universal healthcare and an allocation of patients to specific hospitals based on their living address and pathology. We believe this healthcare model facilitates the creation of multidisciplinary programs since most aspects of care of the same patient can be carried out in the same center. Efficient collaboration of professionals in an institutional approach minimizes fragmentation and variability within treatments and allows rapid coordination of diagnostic and therapeutic procedures in a short preoperative window, as our results show

Good Practice Indicators of proper PBM implementation included the total number and appropriate (according to laboratory-based triggers) RBCT, as well as the number of patients receiving IVI, which all showed a steady improvement over the years of the study. There is little available literature regarding specifically PBM pathways in CCR programs and their compliance or adherence after implementation, and some of them show contrasting results. A network meta-analysis published in 2021 by Roman and Abbasciano [32] did not find important clinical benefits resulting from PBM interventions. However, a recently published study by Shin and Piozzi linked PBM implementation with a reduction of perioperative transfusion and a positive effect on short-term results after surgery, which can also be seen in our results. In our experience, PBM programs must ensure compliance throughout the entire perioperative period and even be integrated with the ERAS programs. There is a lack of evidence supporting this latter statement, but considering its physiological rationale, the implementation of PBM pathways in a CCR program should certainly be facilitated by this, reducing clinical practice variability and providing a cost-effective measure. In our experience, it is possible to follow PBM pathways within an ERAS program in CRC surgery, and it might even facilitate the uptake of fast-track pathways. Future studies should focus on compliance with PBM pathways and guidance for its integration into ERAS programs.

Some limitations of our study include its observational non-controlled design that comes with potential inherent bias, which we, however, believe can nonetheless be an opportunity to appreciate the effect of IVI in routine clinical practice. While it is true that our study lacks a true control group of severely anemic patients without IVI treatment for Group 3 patients, we believe there is value in data obtained from the comparison with Groups 1 and 2 as control groups and also with patients receiving RBC prior to anemia assessment in the PBM Clinic. Their inherent differences and imbalances in baseline characteristics reflect a frequent situation we as clinicians face daily, and we believe our results can shed some light on a difficult decision-making process to avoid variability.

It is also important to note that Hb levels used for the definition of PA and IVI treatment options are still heterogeneous among available literature. This may hamper the extrapolation of our results, but we believe 13 g/dL to be the optimal Hb cutoff value to ensure proper treatment and IVI Carboxymaltose dosed according to a Ganzoni formula to be the best available option for rapid correction of PA on an outpatient basis regime, and both are in accordance with international consensus guidelines.

## 5. Conclusions

Treatment of iron-deficient PA with preoperative IVI administration in CRC patients is a safe and effective measure to significantly increase DOS Hb levels without leading to delays in surgical treatment. Severely anemic patients tend to be older with more comorbidities and appear to especially benefit from preoperative IVI treatment, showing no differences in terms of LOS or severe complications such as anastomotic leak or reintervention compared to less anemic patients. Furthermore, optimal PA assessment and treatment with IVI appear to be more effective than preoperative RBCT and could potentially reduce postoperative RBCT rates in 10% of cases. Compliance with PBM protocols and strict transfusion triggers can be achieved and even improved over time when implemented as part of multidisciplinary institutional policies.

Further studies should focus on studying compliance and adherence to PBM protocols as well as their impact on routine clinical practice and high-volume CRC surgery units.

## Figures and Tables

**Figure 1 cancers-17-00912-f001:**
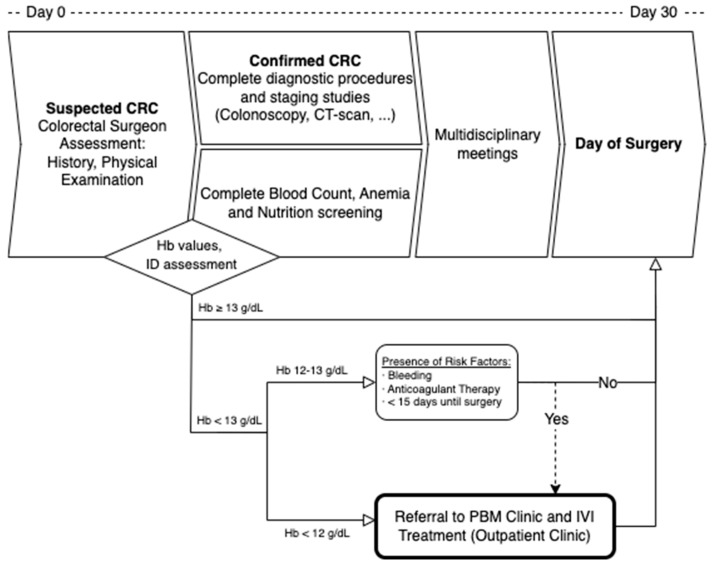
PBM Pathway. Assessment at the PBM Clinic and Administration of IVI Therapy in the Outpatient Anemia Clinic is carried out parallelly to Diagnostic Procedures. Oral Iron and vitamin or folate supplementation (if indicated) are routinely administered but not controlled by the PBM Clinic. CRC: Colorectal Cancer, Hb: Hemoglobin, ID: Iron Deficiency, IVI: Intravenous Iron, CT: Computerized Tomography, PBM: Patient Blood Management.

**Figure 2 cancers-17-00912-f002:**
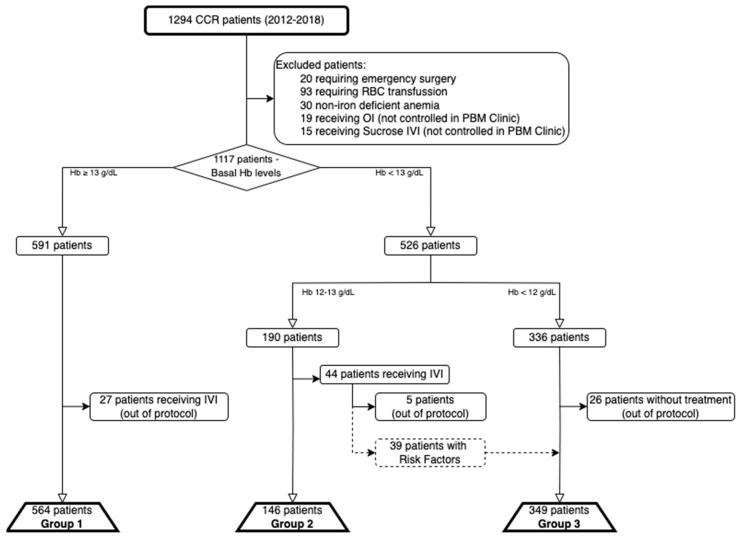
Flow chart of patient selection and group allocation according to Hb levels at diagnosis and treatment received. CRC: Colorectal Cancer, RBC: Red Blood Cell, OI: Oral Iron, IVI: Intravenous Iron, PBM: Patient Blood Management, Hb: Hemoglobin.

**Figure 3 cancers-17-00912-f003:**
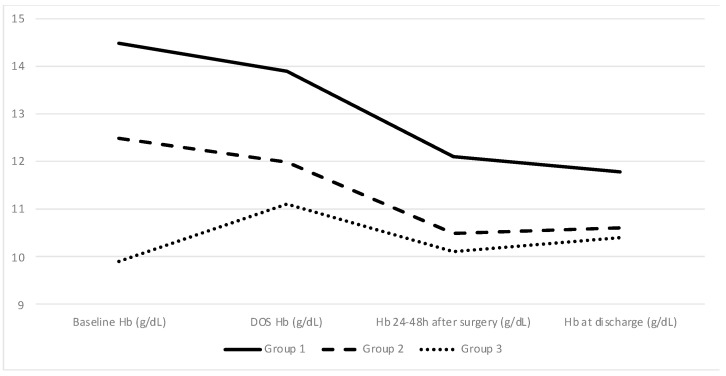
Evolution of Hb levels between diagnosis (Baseline Hb) and DOS. Group 3 values are also a measure of Hb level before and after IVI treatment. Hb: Hemoglobin, DOS: Day Of Surgery.

**Figure 4 cancers-17-00912-f004:**
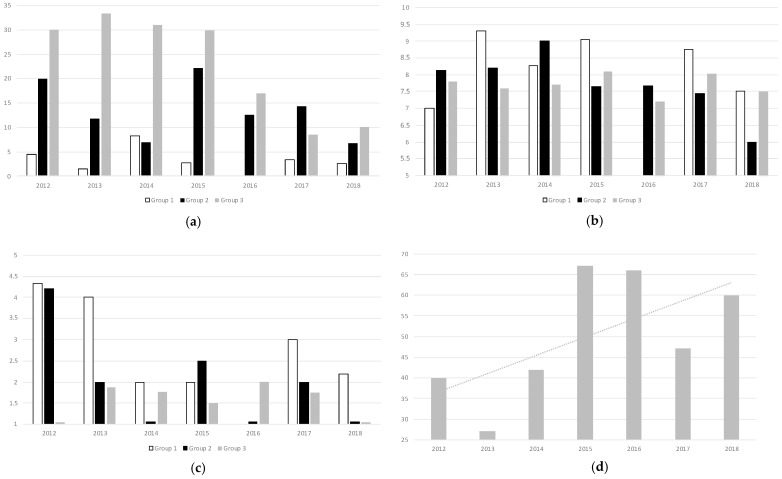
Good Practice Indicators of Compliance with PBM Protocol and Changes Over Time: (**a**) Postoperative RBCT Rate Overview Along the Years of the Study (%); (**b**) Hb Value Trigger Overview Along the Years of the Study (g/dL); (**c**) Number of RBC Units Overview Along the Years of the Study (n); (**d**) Number of Patients receiving IVI Along the Years of the Study (n). RBCT: Red Blood Cell Transfusion, Hb: Hemoglobin, IVI: Intravenous Iron.

**Table 1 cancers-17-00912-t001:** Patients’ baseline characteristics and their significance value for the ANOVA test (normally distributed continuous variables), Kruskal–Wallis (non-normally), and Chi-squared test (categorical variables).

	All Patients (n = 1117)	Comparison Between Groups			
		Group 1 (n = 564)	Group 2 (n = 146)	Group 3 (IVI, n = 349)	*p*-Value
Age, years (m, SD)	70.0 (12.2)	67.3 (11.4)	70.1 (14.3)	74.1 (11.90)	0.00
Sex—M (n, %)	702 (62.9)	398 (70.6)	63 (43.2)	202 (57.9)	0.00
Body Mass Index (BMI), kg/m^2^ (SD)	27.6 (4.6)	27.8 (4.5)	27.8 (4.6)	27.3 (4.5)	0.25
Hypertension (n, %)	644 (57.7)	288 (51.1)	82 (56.2)	237 (67.9)	0.00
Diabetes Mellitus (n, %)	285 (25.5)	564 (18.8)	37 (25.3)	122 (35.0)	0.00
COPD (n, %)	171 (15.3)	83 (14.7)	20 (13.7)	59 (16.7)	0.56
Cardiopathy (n, %)	235 (21.0)	83 (14.7)	36 (24.7)	101 (29.0)	0.00
Anticoagulant therapy (n, %)	94 (8.4)	27 (4.8)	14 (9.6)	43 (12.3)	0.00
ASA score (n, %)					
ASA I	36 (3.2)	26 (4.6)	5 (3.4)	4 (1.2)	0.00
ASA II	805 (72.1)	441 (78.2)	106 (72.6)	218 (62.5)	0.00
ASA III	270 (24.2)	95 (16.8)	33 (22.6)	125 (35.8)	0.00
ASA IV	6 (0.5)	2 (0.4)	2 (1.4)	2 (0.6)	0.00
Pathologic Stage (n, %)					
Stage 0	42 (3.8)	31 (5.6)	5 (3.5)	5 (1.5)	0.00
Stage I	210 (19.1)	126 (22.6)	28 (19.7)	46 (13.5)	0.00
Stage II	406 (37.0)	184 (33.0)	53 (37.3)	150 (44.0)	0.00
Stage III	375 (34.2)	186 (33.4)	45 (31.7)	117 (34.3)	0.00
Stage IV	65 (5.9)	30 (5.4)	11 (7.8)	23 (6.7)	0.90
Neoadjuvant therapy (n, %)	233 (20.9)	132 (23.4)	39 (26.7)	47 (13.5)	0.00

COPD: Chronic Obstructive Pulmonary Disease, ASA: American Society of Anesthesiology.

**Table 2 cancers-17-00912-t002:** Overview of anemia values and their significance value for the ANOVA test (normally distributed continuous variables), Kruskal–Wallis (non-normally), and Chi-squared test (categorical variables).

	Comparison Between Groups			
	Group 1 (n = 564)	Group 2 (n = 146)	Group 3 (IVI, n = 349)	*p*-Value
Baseline Hb, g/dL (m, SD)	14.5 (1.0)	12.5 (0.3)	9.9 (1.5)	0.00
Ferritin, µg/L (md, IR)	65 [29.1–127.6]	25.3 [15–54]	14.5 [7–42.2]	0.00
Transferrin Saturation, % (md, IR)	19 [13.6–25]	13.3 [8.5–18]	6.8 [4–13.1]	0.00
Time Diagnosis-Surgery, days (md, IR)	30 [21.50]	31.5 [19.51]	31 [22.48]	0.08
Hb DOS, g/dL (m, SD)	13.9 (1.3)	12.0 (1.0)	11.1 (1.5)	0.00
Hb 24–48 after surgery, g/dL (m, SD)	12.1 (1.5)	10.5 (1.1)	10.1 (1.3)	0.00
Hb at discharge, g/dL (m, SD)	11.8 (1.6)	10.5 (1.1)	10.4 (1.3)	0.00

Hb: Hemoglobin, Time Diagnosis-Surgery: Days between available Baseline Hb and DOS; Hb DOS: Day Of Surgery value of Hb.

**Table 3 cancers-17-00912-t003:** Surgical results and postoperative complications and their significance value for the ANOVA test (normally distributed continuous variables), Kruskal–Wallis (non-normally), and Chi-squared test (categorical variables).

	All Patients (n = 1117)	Comparison Between Groups			
		Group 1 (n = 564)	Group 2 (n = 146)	Group 3 (IVI, n = 349)	p-Value
Laparoscopy (n, %)	665 (59.4)	360 (63.8)	73 (50)	193 (55.3)	0.00
Conversion (n, %)	65 (9.9)	31 (8.6)	9 (12.33)	22 (11.58)	0.43
Type of surgery (n, %)					
Right colectomy	341 (30.5)	114 (20.2)	38 (26.0)	165 (47.3)	0.00
Left colectomy	92 (8.2)	51 (9.0)	12 (8.2)	25 (7.2)	0.06
Sigmoidectomy	318 (28.5)	184 (32.6)	43 (29.5)	82 (23.5)	0.00
Low Anterior Resection	286 (25.6)	175 (31.0)	43 (29.5)	50 (14.3)	0.00
Miles procedure	55 (4.92)	31 (5.5)	5 (3.4)	16 (4.6)	0.57
Subtotal colectomy	16 (1.4)	7 (1.2)	3 (2.1)	6 (1.7)	0.29
Hartmann procedure	9 (0.8)	2 (0.4)	2 (1.4)	5 (1.4)	0.52
Clavien-Dindo I-II (n, %)	273 (24.4)	133 (23.6)	29 (19.9)	95 (27.2)	0.02
Surgical Site Infection	79 (7.1)	40 (7.1)	9 (6.2)	27 (7.7)	0.82
Intraabdominal abscess	21 (1.9)	10 (1.8)	2 (1.4)	7 (2.0)	0.88
Minor anastomotic leak	51 (4.6)	31 (5.5)	8 (4.8)	16 (4.6)	0.06
Low Gastrointestinal Bleeding	20 (1.8)	10 (1.8)	2 (1.4)	7 (2.0)	0.88
Paralytic ileus	198 (17.7)	90 (16.0)	25 (17.1)	73 (21.0)	0.16
Clavien-Dindo III-IV (n, %)	72 (6.5)	24 (4.3)	10 (6.9)	32 (9.2)	0.02
Major anastomotic leak	31 (2.8)	16 (2.8)	2 (1.4)	12 (3.5)	0.40
Hemoperitoneum	12 (1.1)	4 (0.7)	1 (0.7)	7 (2.0)	0.19
Respiratory Failure	23 (2.1)	4 (0.7)	6 (4.1)	11 (3.2)	0.00
Heart Failure	18 (1.6)	3 (0.5)	1 (0.7)	11 (3.2)	0.01
RBCT (n, %)	121 (10.9)	19 (3.4)	19 (13.0)	75 (21.6)	0.00
Mortality (n, %)	15 (1.4)	5 (0.9)	2 (1.4)	6 (1.7)	0.54
LOS, days (m, SD)	9.2 (6.4)	8.8 (6.0)	9.0 (6.2)	9.8 (7.0)	0.07

RBCT: Red Blood Cell Transfusion, LOS: Length of Stay.

## Data Availability

Data is available upon request.

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
