# Peer review of "Postoperative Results After Patient Blood Management with Intravenous Iron Treatment Implementation for Preoperative Anemia: Prospective Cohort Study of 1294 Colorectal Cancer Patients"

_cancers, 2025, doi:10.3390/cancers17060912_

Round 1
Reviewer 1 Report
Comments and Suggestions for Authors
The scope of the article and the importance of the subject matter is clearly stated in the introduction. The topic is interesting and is very useful because the results showed a decrease in the need of postoperative blood transfusions in patients who received IVI. The article also features a variety of tables and the information is presented clearly and in a manner that is easy to understand. The methods and results are presented clearly and in detail. Overall, a very useful article which provides interesting insight about the treatment of preoperative anemia with IVI.
Reviewer 2 Report
Comments and Suggestions for Authors
1. This study provides initial evidence of IVI’s feasibility in colorectal cancer patients with preoperative anemia, but it does not fully establish its superiority or clinical value. This is a single-center, prospective observational cohort study, which inherently cannot completely eliminate selection bias and confounding factors. Since patients were not randomly assigned to different treatment groups, imbalances in baseline characteristics could have influenced the results.
2. Although the study compared non-anemic patients (Group 1), mildly anemic patients (Group 2), and anemic patients receiving IVI (Group 3), it lacks a true control group of anemic patients who did not receive any anemia treatment. This omission makes it difficult to determine whether factors other than IVI (such as ERAS programs) contributed to the postoperative outcomes.
3. The threshold for defining anemia varies across international guidelines, potentially affecting the external validity of the findings, and IVI dosing was determined using a modified Ganzoni formula or a simplified strategy, but the study did not compare these methods, which might influence the consistency of treatment outcomes.
4. In Group 3 (IVI-treated patients), Hb increased from 9.9 g/dL to 11.1 g/dL. While statistically significant (p < 0.001), the clinical impact of this modest increase remains questionable in terms of improving postoperative outcomes, and different patients had varying transfusion thresholds, which might affect the true effect of IVI in reducing RBCT use.
5. Did IVI Truly Reduce RBCT Use? The study reports that the postoperative RBCT rate was 21.6% in the IVI group, lower than 32.6% in patients who received preoperative RBCT. However, the baseline Hb levels of these two groups were different, introducing potential bias.
6. The IVI group and the mild anemia group (Group 2) showed no significant difference in discharge Hb levels (p = 0.265), raising concerns about whether IVI provides additional benefit.
7. The study states that Clavien-Dindo III-IV (severe complications) did not differ significantly among the groups, yet the IVI group had higher rates of respiratory and cardiovascular complications, and no further analysis was conducted to determine whether IVI increased fluid overload and contributed to cardiovascular complications, which is a known potential risk of IVI treatment.
8. The study was conducted in a single hospital within Spain’s public healthcare system, where resource allocation and patient management may not be generalizable to other countries or private healthcare settings.
9. IVI is generally more expensive than RBCT, yet the study does not provide a cost-effectiveness comparison, which is crucial for clinical decision-making and reimbursement policies.
10. The study only evaluates 30-day postoperative outcomes and does not assess long-term survival, cancer recurrence, or persistent anemia, making it difficult to determine IVI’s long-term clinical benefits.
Comments on the Quality of English LanguageThe English could be improved to more clearly express the research.
Reviewer 3 Report
Comments and Suggestions for Authors
The authors showed that aggressive iron therapy for preoperative iron deficiency anemia in advanced colorectal cancer improved Hb levels and maintained them from the time of surgery until discharge. This study is a large cohort study and is important in managing anemia. The authors have several issues that need to be resolved before publication.
Major concerns
- In the authors' study, was the improvement of iron deficiency anemia due to preoperative iron therapy before surgery the reason why Hb levels were maintained from postoperative to discharge in Gruop3? Do authors use iron therapy postoperatively?
- In this study, Group 3 is being treated with aggressive therapeutic intervention with iron medication. Does this study show ferritin and transferrin levels immediately before surgery? Showing how much these values have improved makes iron therapy even more significant.
- In the results of this study, there is a detailed description of 99 patients who had RBCT before surgery. What was the reason for performing RBCT in this population, unlike Group 3, instead of treatment with iron before surgery?
Round 2
Reviewer 2 Report
Comments and Suggestions for Authors
The author has revised following the suggested revisions.
Comments on the Quality of English LanguageYou can improve your English proficiency to express research findings more clearly.
Reviewer 3 Report
Comments and Suggestions for Authors
The presented manuscript is revised adequately. Thank you for this study, which has shown the effectiveness and safety of preoperative iron administration.